

# Transcriptome analysis of green and purple fruited pepper provides insight into novel regulatory genes in anthocyanin biosynthesis

Huaqiang Tan[1], Liping Li[1], Manman Tie[2], Ronghai Lu[1], Shaokun Pan[1] and Youwan Tang[1]

[1] Chengdu Academy of Agriculture and Forestry Sciences, Chengdu, Sichuan, China
[2] Agricultural and Rural Bureau of Lushan County, Yaan, Sichuan, China

Corresponding author
Youwan Tang, 2919792007@qq.com

## ABSTRACT

**Background**. Pepper (*Capsicum annuum* L.) is a valuable horticultural crop with economic significance, and its purple fruit color is attributed to anthocyanin, a phytonutrient known for its health-promoting benefits. However, the mechanisms regulating anthocyanin biosynthesis in pepper have yet to be fully elucidated.

**Methods**. RNA sequencing (RNA-seq) was utilized to analyze the transcriptome of fruits from three purple-fruited varieties (HN191, HN192, and HN005) and one green-fruited variety (EJT) at various developmental stages. To determine the relationships between samples, Pearson correlation coefficients (PCC) and principal component analysis (PCA) were calculated. Differential expression analysis was performed using the DESeq2 package to identify genes that were expressed differently between two samples. Transcription factors (TF) were predicted using the iTAK program. Heatmaps of selected genes were generated using Tbtools software.

**Results**. The unripe fruits of HN191, HN192, and HN005, at the stages of 10, 20, and 30 days after anthesis (DAA), display a purple color, whereas the unripe fruits of variety EJT remain green. To understand the molecular basis of this color difference, five transcriptome comparisons between green and purple fruits were conducted: HN191-10 *vs* EJT-10, HN191-20 *vs* EJT-20, HN191-30 *vs* EJT-30, HN192-30 *vs* EJT-30, and HN005-30 *vs* EJT-30. Through this analysis, 503 common differentially expressed genes (DEGs) were identified. Among these DEGs, eight structural genes related to the anthocyanin biosynthesis pathway and 24 transcription factors (TFs) were detected. Notably, one structural gene (MSTRG.12525) and three TFs (T459_25295, T459_06113, T459_26036) exhibited expression patterns that suggest they may be novel candidate genes involved in anthocyanin biosynthesis. These results provide new insights into the regulation of anthocyanin biosynthesis in purple pepper fruit and suggest potential candidate genes for future genetic improvement of pepper germplasm with enhanced anthocyanin accumulation.

## INTRODUCTION

Anthocyanins are natural pigments responsible for a wide range of colors—from orange and red to purple and blue—in flowers, seeds, fruits, and vegetative tissues (*Tanaka, Sasaki & Ohmiya, 2008*). These flavonoids have important biological functions in plants, such as attracting insects to spread seeds, protecting against photo-oxidative damage, and improving resistance to stresses (*Gould, 2004*; *Steyn et al., 2002*; *Chalker-Scott, 1999*). More importantly, anthocyanins have antioxidant properties that benefit human health and may help prevent chronic and degenerative illnesses like cancer and cardiovascular disease (*Reddy, Alexander-Lindo & Nair, 2005*; *Lamy et al., 2007*; *De Pascual-Teresa, Moreno & García-Viguera, 2010*). As a result, the biosynthesis of anthocyanin becomes a focus of interest for the genetic improvement of crops.

The biosynthesis of anthocyanin has been extensively studied in various plants and is a branch pathway of flavonoid metabolism, involving two groups of genes: structural and regulatory (*Gonzali, Mazzucato & Perata, 2009*). Structural genes encode enzymes that directly participate in anthocyanin biosynthesis, such as chalcone synthase (CHS), chalcone isomerase (CHI), flavanone 3-hydroxylase (F3H), flavonoid 3′5′-hydroxylase (F3′5′H), dihydroflavonol 4-reductase (DFR), anthocyanidin synthase (ANS), and flavonoid 3-O-glucosyltransferase (UFGT) (*Holton & Cornish, 1995*; *Zhang et al., 2015*). *CHS*, *CHI* and *F3H* are categorized as early biosynthetic genes (EBGs), while *F3′H*, *F3′5′H*, *DFR*, *ANS*, and *UFGT* are classified as late biosynthetic genes (LBGs) (*Chen et al., 2022*). Meanwhile, regulatory genes encode transcription factors (TFs) that control the expression of structural genes. Three main types of TFs have been identified, including my elob lastosis (MYB), basic helix-loop-helix (bHLH), and WD40-repeat proteins (*Zhang, Butelli & Martin, 2014*), which can form a regulatory complex called MYB-bHLH-WD40 (MBW) and bind to the promoters of structural genes, especially LBGs, leading to their transcriptional activation and expression (*Stommel et al., 2009*; *Petroni & Tonelli, 2011*).

Pepper (*Capsicum annuum* L.) is among the most commonly cultivated vegetable crops worldwide. The unripe pepper fruit can exhibit various colors such as dark green, green, light green, purple, yellow, or ivory (*Liu et al., 2020b*). The purple color arises from the accumulation of anthocyanins (*Lightbourn et al., 2008*). It was initially reported that a gene called gene *A* is responsible for the purple fruit color in pepper (*Peterson, 1959*). This gene *A* was subsequently isolated from pepper and found to encode an R2R3 MYB transcription factor similar to Anthocyanin2 in Petunia (*Borovsky et al., 2004*). Overexpression of this *CaAn2* gene in *arabidopsis* and tobacco plants resulted in a purple phenotype with increased anthocyanin content (*Jung et al., 2019*). Further investigation revealed that a non-long terminal repeat (non-LTR) retrotransposon was inserted in the promoter region of *CaAn2* in purple *C. annuum* plants. This retrotransposon may activate the expression of *CaAn2* by recruiting transcription factors at the 3′ UTR (*Jung et al., 2019*). Additionally, genetic mapping and transcriptome research have aimed to identify more genes involved in anthocyanin biosynthesis. *Liu et al. (2020a)* employed a map-based cloning strategy to fine-map a structural gene (*Capana10g001978*) responsible for anthocyanin biosynthesis, while *Wang et al. (2018)* utilized a combination of specific length amplified fragment

sequencing (SLAF-seq) and bulked segregant analysis (BSA) to identify 12 candidate genes associated with anthocyanin accumulation. Furthermore, transcriptome analysis identified several genes, including *CaANT1*, *CaANT2*, *CaAN1*, and *CaTTG1*, involved in anthocyanin accumulation in the purple pepper fruit variety No. Co62 (*Tang et al., 2020*). A combined analysis of metabolome and transcriptome identified ten highly expressed genes encoding transcription factors in the purple fruit variety C1-P, including 2 WD (WD68 and WD44), 1 bHLH (bHLH143), and 1 MADS-box protein (AGL16) (*Liu et al., 2020b*). Moreover, *Meng et al. (2022)* identified 59 unigenes, including 7 enzymes and 8 transcription factors, as candidate genes involved in anthocyanin biosynthesis in the purple fruit varieties L29 and L66.

Due to their attractive color, high antioxidant capacity, and positive effects on shelf-life, there is an increasing interest in uncovering the mechanism of anthocyanin metabolism in Solanaceous vegetables such as pepper, eggplant, tomato and potato. Current knowledge on anthocyanins in the Solanaceous vegetables has been reviewed, including biochemistry and biological function of anthocyanins, as well as their genetic and environmental regulation (*Liu et al., 2018b*). Despite the abundance of literature on anthocyanins in Solanaceae, the transcriptional regulation of anthocyanin biosynthesis in Capsicum remains to be clearly elucidated (*Tang et al., 2020*). The aim of this study was to investigate the transcriptome dynamics of three purple-fruited pepper varieties (HN191, HN191, HN005) and a green-fruited variety (EJT) across different stages of development using RNA-seq, thus gaining further insight into the genes and regulatory networks involved in anthocyanin biosynthesis in purple pepper fruit.

## MATERIALS & METHODS

### Plant materials

Four pepper varieties, HN191, HN192, HN005 and EJT were grown in the experimental field of Chengdu Academy of Agriculture and Forestry Sciences (Chengdu, Sichuan, China). Erjingtiao (EJT) is a famous local pepper variety in Sichuan Province with a long cultivation history. It is characterized by its thin skin, thick flesh, delightful fragrance, moderate spiciness, and high nutritional value. This pepper is an essential ingredient in authentic Sichuan cuisine. The unripe fruits are green, while the ripe ones turn red. It is a very representative green fruit pepper material. HN191, HN192, and HN005 are inbred lines developed by our research group. These lines exhibit purple unripe fruits that transform into red mature fruits. In 2020, they were officially recognized by the Sichuan Provincial Non-Major Crops Committee.

Fruits from HN191 and EJT were harvested at 10, 20, 30, and 60 DAA, while fruits from HN192 and HN005 were harvested at 30 and 60 days after anthesis (DAA). For each variety and developmental stage, we collected three biological replicates and stored them at −80 °C for further experiments.

### Assessment of anthocyanin

Anthocyanin mixtures from 30 DAA fruits of four pepper varieties (HN191, HN192, HN005, and EJT, respectively) were extracted and measured according to the agricultural

standard of China (NY/T 2640-2014). The anthocyanin standard included delphinidin, cyanidin, petunidin, pelargonidin, peonidin, and malvidin, which were dissolved in methanol with 10% hydrochloric acid.

## Microscope observation
Fresh fruits of HN191 at 30 DAA were cut to create free-hand sections and observed under a microscope (BX41; Olympus, Tokyo, Japan).

## RNA extraction and sequencing
The pepper fruit samples that had been stored were removed from the −80 °C ultra-low temperature freezer and promptly placed into a foam box containing liquid nitrogen. Subsequently, the samples were taken out from the foam box and ground into powder using sterilized mortar and pestle in liquid nitrogen. The powdered samples were then transferred into 1.5 ml RNase-free microtubes (Corning Incorporated, Corning, NY, USA) immediately. Afterwards, total RNA was extracted using the RNAprep Pure extraction kit (Tiangen Biotech Co, Ltd., Beijing, China) in accordance with the manufacturer's instructions. The purity and integrity of the RNA samples were evaluated using the Agilent Bioanalyzer 2100 system (Agilent Co., Ltd., Beijing, China). The sequencing library of all samples were constructed, and sequenced on the DNBSEQ-T7 platform in China National GeneBank (CNGB).

## Data analysis
The adaptor and low-quality reads were first removed using Fastp (*Chen et al., 2018*). The resulting clean reads were then mapped to the pepper genome (Genbank assembly accession: GCA_000512255.2) using HISAT2 software (*Kim, Langmead & Salzberg, 2015*; *Kim et al., 2014*). To calculate the FPKM (fragments per kilobase of transcript per million mapped reads) for each gene, StringTie and Ballgown software were employed (*Pertea et al., 2016*).

The correlation between different samples was analyzed using the Pearson Correlation Coefficient (PCC) and Principal Component Analysis (PCA), which were implemented in the Factoextra and FactoMineR packages in R software.

For the differential expression analysis between samples, DESeq2 software was utilized (*Varet et al., 2016*). Genes with |log2Fold Change| $\geq$ 1 and a False Discovery Rate (FDR) < 0.05 were considered as differentially expressed genes (DEGs). The iTAK program was used to predict transcription factors from the DEGs (*Zheng et al., 2016*). Additionally, heatmaps of selected genes were generated using Tbtools software (*Chen et al., 2020*).

## qRT-PCR analysis
The qPCR assay was configured following the recommendations of 'The MIQE guidelines' (*Bustin et al., 2009*). Ten structural genes and eleven regulatory genes (transcription factors, TFs) were selected to validate the RNA-seq results. Primers for qRT-PCR were designed using the Primer3 online software (http://bioinfo.ut.ee/primer3-0.4.0/) and synthesized by Sangon Biotech Co., Ltd., Shanghai, China (Table S1). For the cDNA synthesis, 1 µg of total RNA was reverse transcribed using the PrimeScript™ RT reagent Kit (Takara Bio

Inc., San Jose, CA, USA), following the manufacturer's protocol. Quantitative RT-PCR was performed on a CFX96 Real-Time PCR system (Bio-Rad Laboratories Inc., Hercules, CA, USA) using TB Green® Premix Ex Taq™ II (Takara Bio Inc., Beijing, China). The consumables used include RNase-free tips and 8 Strip PCR tubes from Axygen® Brand Products (Corning Incorporated, Corning, NY, USA). The quantification was performed in triplicate using 25 μL reactions. Each reaction included 12.5 μl of TB Green Premix Ex Taq II, 1 μl of each primer (10 μM), 8.5 μl RNase-free water, and 2 μl of 1:5 diluted cDNA. The PCR amplification conditions consisted of an initial denaturation at 95 °C for 30 s, followed by 40 cycles of denaturation at 95 ° C for 5 s and annealing at 60 °C for 30 s. A melting curve was obtained at the end of each PCR by gradually increasing the temperature to 95 °C (increment rates of 0.5 °C/s) after cooling to 65 °C for 5 s. Each gene was analyzed on the same amplification for all samples, so inter-run calibration was not necessary. The data obtained were analyzed by the Bio-Rad CFX Manager software (version 3.0), which generated the raw quantification cycle (Cq) values for each reaction. The relative expression of the selected genes was normalized to a pepper actin gene (GenBank ID: T459_30033) and calculated using the 2-$\Delta\Delta$CT method. Further qPCR details are supplied in a MIQE checklist table (Table S2).

## RESULTS

### Phenotype analysis

In this study, four pepper varieties (HN191, HN192, HN005, and EJT) were selected as research materials. The unripe fruits of HN191, HN192, and HN005 have a purple color at 10, 20, and 30 days after anthesis (DAA), while the unripe fruits of EJT are green (Fig. 1). However, all four varieties have red ripe fruits at 60 DAA. The result of HPLC analysis showed that delphinidin was the only anthocyanin present in the three purple-fruited varieties (HN191, HN192, and HN005), while the levels of cyanidin, petunidin, pelargonidin, peonidin, and malvidin were below the detection threshold. No anthocyanins were detected in the green-fruited variety EJT. Furthermore, when we examined the fruits under a microscope, we observed that anthocyanins were only accumulated in the exocarp of HN191 fruits at 30 DAA (Fig. 2).

### Global transcriptome results

We performed RNA-seq to investigate the dynamics of the transcriptome in fruit samples obtained from HN191 and EJT at four stages (10, 20, 30, and 60 DAA), as well as fruit samples from HN192 and HN005 at two stages (30 and 60 DAA). We analyzed three independent biological replicates of fruits for each variety and stage, resulting in a total of 36 samples. The extracted RNA exhibited a yield ranging from 14 to 52 μg and an RNA Integrity Number (RIN) ranging from 7.2 to 9.1, indicating relatively high RNA quality (Table S3).

A total of 7.25 billion clean reads, representing 725.48 GB of clean bases, were generated from all samples (Table S4) and were mapped to the pepper genome (Kim et al., 2014) at an average rate of 94.03% (range: 92.80%–94.80%). Power analysis on the transcriptome data was conducted using the RNASeqPower package (Hart et al., 2013). Based on a sample

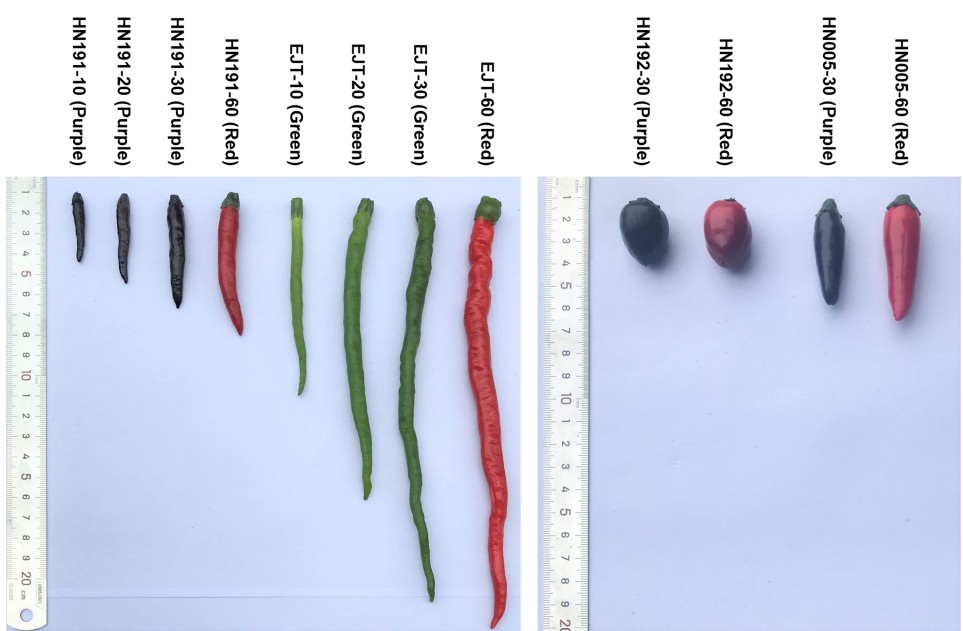

**Figure 1** Fruit samples of four pepper varieties at different developmental stages.

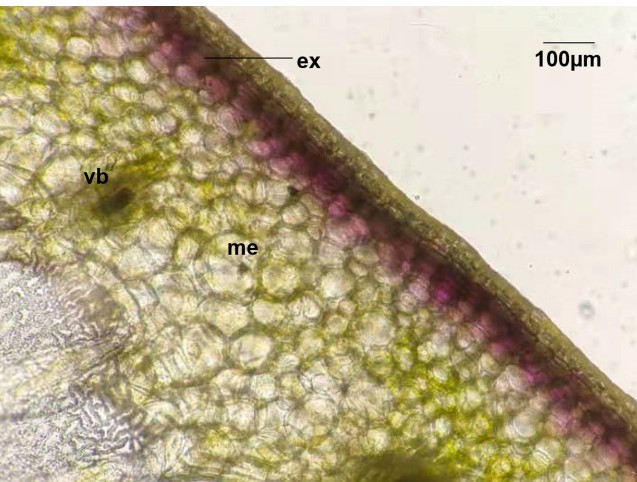

**Figure 2** Distribution of Anthocyanins in HN191 fruits at 30 days after anthesis (DAA). Me, mesocarp; Vb, vascular bundle; Ex, exocarp.

size of three (the number of biological replicates used in this study), the statistical power among different groups ranged from 0.85 to 0.91, with an average value of 0.87 (Table S5).

For each gene, the FPKM values were calculated, and genes with FPKM >1 were considered expressed. In EJT and HN191 at 10, 20, 30, and 60 DAA, as well as HN192 and HN005 at 30 and 60 DAA, a total of 19,303–21,826 genes were found to be expressed (Fig. 3A). The expression of 9%–11% of genes was very high (FPKM > 20) in the four

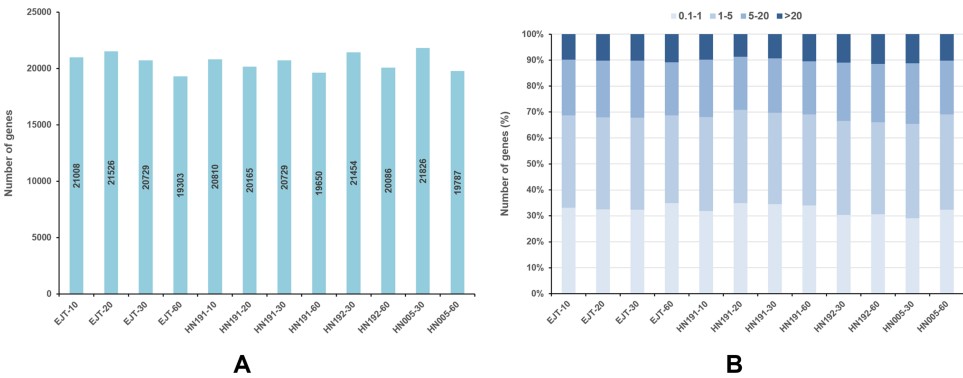

**Figure 3** **Genes expressed in four pepper varieties at different developmental stages.** (A) Genes expressed with FPKM > 1. (B) The fraction of genes with differential expression in EJT and HN191 at 10, 20, 30, and 60 DAA, as well as HN192 and HN005 at 30 and 60 DAA.

pepper varieties at different developmental stages (Fig. 3B). Genes with high expression (FPKM 5 to $\leq$ 20), moderate expression (FPKM 1 to $\leq$ 5), and low expression (FPKM 0.1 to $\leq$ 1) accounted for 20%–23%, 34%–37%, and 29%–35% of the total genes, respectively.

## Comparison of global transcriptome between samples

The global differences in transcriptomes during fruit development in HN191, HN192, HN005, and EJT varieties were investigated. The mean FPKM values of three biological replicates were used to calculate the Pearson correlation coefficients (PCC) for all samples (Fig. 4A). The PCC values between purple and green unripe fruits ranged from 0.69 to 0.88 (HN191-10 *VS* EJT-10, 0.88; HN191-20 *VS* EJT-20, 0.80; HN191-30 *VS* EJT-30, 0.69; HN192-30 *VS* EJT-30, 0.88; HN005-30 *VS* EJT-30, 0.88). However, for each variety, the PCC values between unripe fruits (10, 20, 30 DAA) and ripe fruits (60 DAA) were relatively low, measuring below 0.6.

Additionally, principal component analysis (PCA) was performed (Fig. 4B). The distances between samples in the plot indicate the similarity of their transcriptional programs. Samples of unripe fruits, both green and purple, were located on the left side of the plot, indicating that the transcriptional activity of purple and green unripe fruits is similar despite the difference in fruit color. On the other hand, samples of ripe fruits clustered on the right side, suggesting that unripe and ripe fruits are more likely to exhibit different transcriptomes and functions/activities, which aligns with the Pearson correlation coefficient results.

## DEG analysis

In order to investigate genes associated with anthocyanin biosynthesis in pepper, our focus was on analyzing the differentially expressed genes (DEGs) between five comparisons of purple and green fruit varieties at 10, 20, and 30 days after anthesis (DAA). These comparisons included HN191 and EJT at 10 DAA, HN191 and EJT at 20 DAA, HN191 and EJT at 30 DAA, HN192 and EJT at 30 DAA, and HN005 and EJT at 30 DAA.

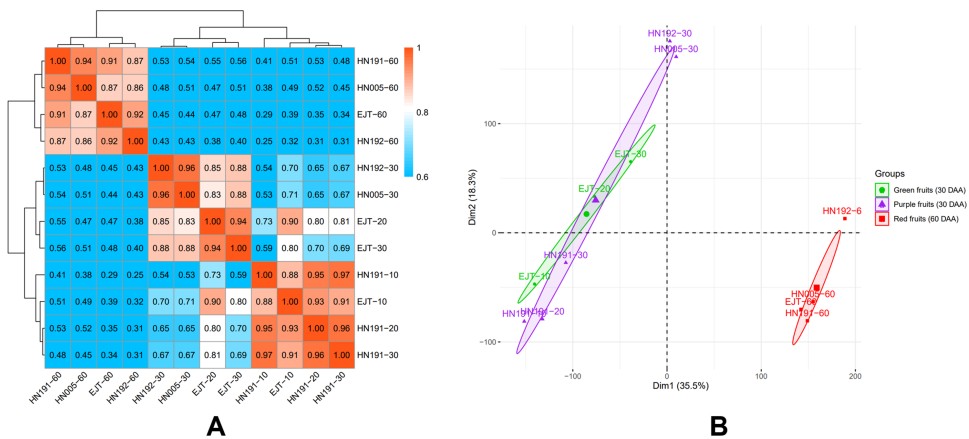

**Figure 4  Correlation between transcriptomes of four pepper varieties at different developmental stages.** (A) The results of Pearson correlation coefficient analysis. (B) The results of principal component analysis. Ellipses indicate 95% confidence intervals.

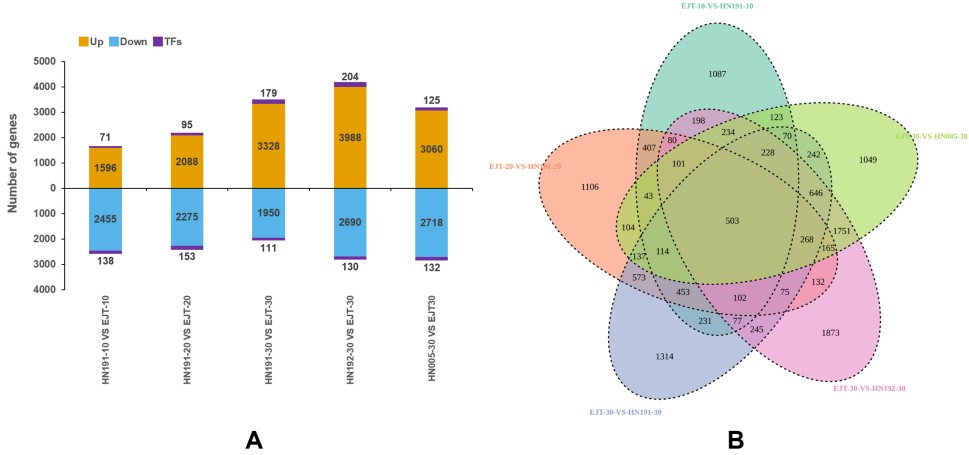

**Figure 5  Statistics of differential gene expression in purple fruit varieties (HN191, HN192, HN005) compared with the green fruit variety (EJT) at different stages.** (A) The statistics of diûerentially expressed genes. Orange, blue, and purple indicate upregulated genes, downregulated genes, and transcription factors (TFs), respectively. (B) Venn plot of differentially expressed genes between the five comparisons.

In total, we identified 14,734 upregulated genes, which included 674 genes encoding transcription factors (TFs), and 12,752 downregulated genes, which included 664 TF-encoding genes (Fig. 5A). The highest number of differentially expressed genes (7,012) was observed between HN192 and EJT at 30 DAA (HN192-30 *VS* EJT-30), followed by HN005 and EJT at 30 DAA (HN005-30 *VS* EJT-30). Conversely, the smallest number of differentially expressed genes (4,260) was found between HN191 and EJT at 10 DAA (HN191-10 *VS* EJT-10).

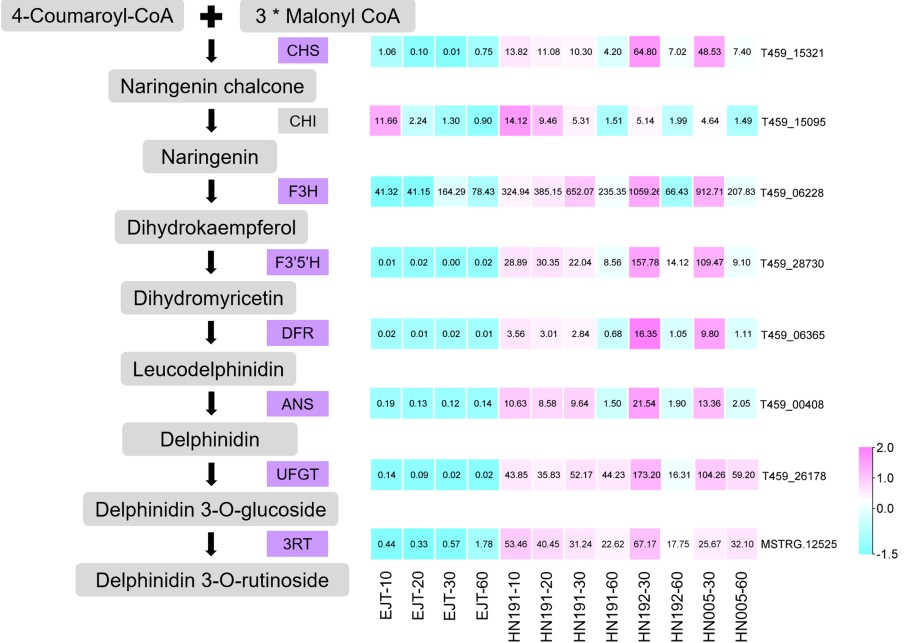

**Figure 6** Structural genes of anthocyanin biosynthesis identified from common DEGs.

## Common DEGs between purple and green fruits

The key distinguishing factor among these five comparisons is the fruit color, with green fruits being compared to purple fruits. Venn diagrams of the DEGs in each of the five comparisons were plotted using Venny2.1 (Fig. 5B). Among these comparisons, a total of 503 DEGs were identified that were common across all five, indicating their potential involvement in the biosynthesis of anthocyanin in pepper fruit.

## Structure genes identified from common DEGs

Structural genes encode enzymes that directly participate in the biosynthesis of anthocyanin. From the pool of 503 common genes, we identified 7 structural genes involved in the anthocyanin biosynthesis pathway, namely *CHS* (T459_15321), *F3H* (T459_06228), *F3′5′H* (T459_28730), *DFR* (T459_06365), *ANS* (T459_00408), *UFGT* (T459_26178), and *3RT* (MSTRG.12525), as shown in Fig. 6. The expression levels of these genes were relatively higher in purple fruits (HN191 at 10, 20, 30DAA, HN192 at 30 DAA, HN005 at 30DAA) compared to green fruits (EJT at 10, 20, 30 DAA). Moreover, their expression in purple fruits sharply decreased at 60 DAA, aligning with the observed pattern of anthocyanin accumulation. An additional important structural gene, *CHI* (T459_15095), was not part of the pool of 503 common DEGs because its fold change between HN191 and EJT at 10 DAA did not meet the threshold of 2. Nevertheless, in the remaining four comparisons, the expression levels of *CHI* in purple fruits were more than two times higher than in green fruits. For further reference, the sequences of these 8 structural genes can be found in Table S6.

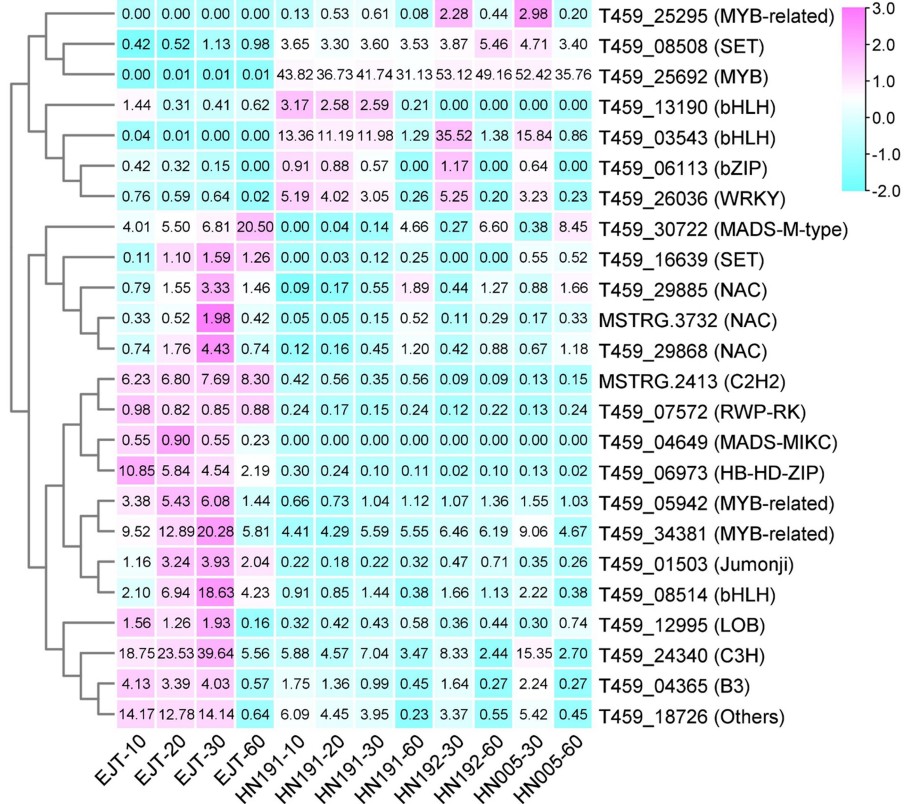

**Figure 7** Heatmap of transcription factors (TFs) predicted from common DEGs.

## TF prediction from common DEGs

Regulatory genes are responsible for encoding transcription factors that play a crucial role in the biosynthesis of anthocyanin (*Lloyd et al., 2017*). To identify these transcription factors, we utilized the iTAK software to predict them from the 503 common DEGs. As a result, a total of 24 transcription factors were identified (Table S7). We then generated a heatmap (Fig. 7) based on the FPKM values of these 24 transcription factors. Based on their expression patterns, these transcription factors can be divided into two distinct groups. The first group consisted of seven transcription factors (from T459_25295 to T459_26036) that exhibited relatively low expression in green fruits (EJT at 10, 20, 30 DAA), but showed higher expression in fruits of the three purple varieties (HN191 at 10, 20, 30 DAA, HN192 at 30 DAA, HN005 at 30 DAA). Conversely, the second group included 17 transcription factors (from T459_30722 to T459_18726) that displayed the opposite expression pattern. These findings suggest that the seven transcription factors in the first group may be closely associated with the biosynthesis of anthocyanin in pepper fruit.

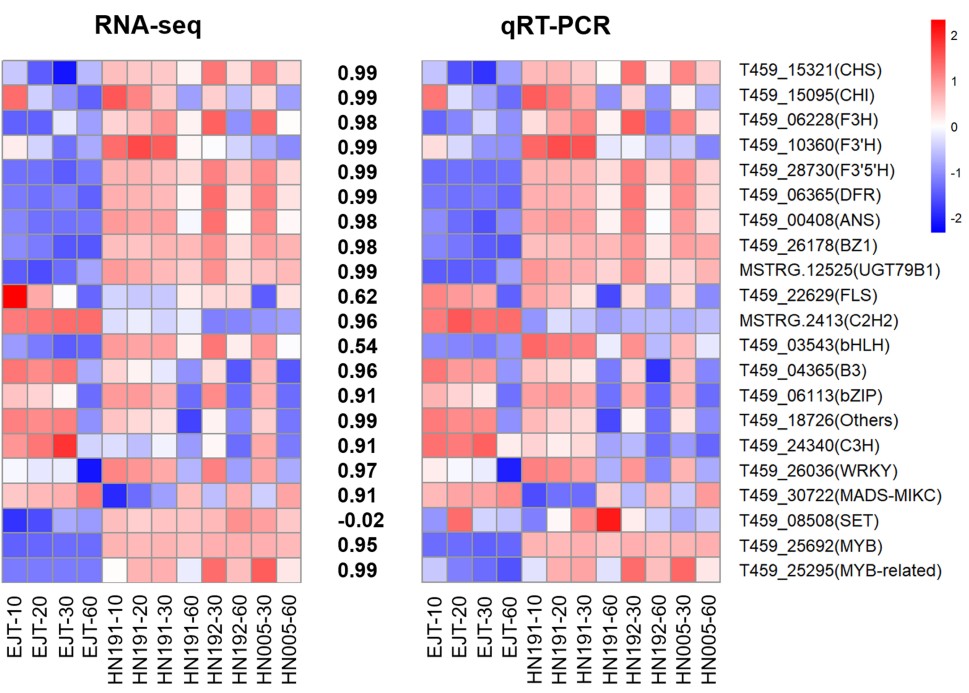

**Figure 8 Validation of expression levels of selected genes in four pepper varieties.** The heatmaps represent expression profiles of selected genes (labeled on the right side). The left heatmap is derived from RNA-seq analysis, and the right heatmap is derived from qRT-PCR analysis. The color scale represents the Z-score. The values between the two heatmaps represent the correlation coefficient between the expression data of selected genes obtained from RNA-seq and qRT-PCR analyses.

## qPCR validation

To validate the reliability of the RNA-seq data, we selected and analyzed 10 structural genes and 11 regulatory genes (transcription factors) associated with anthocyanin biosynthesis using qPCR.

The qPCR results demonstrated that the expression patterns of these selected genes were largely consistent with those observed in the RNA-seq data, except for one gene T459_08508 (Fig. 8, Table S8, and Table S9). The Pearson correlation coefficient between the RNA-seq and qRT-PCR data ranged from 0.54 to 0.99, with an average value of 0.88. This suggests that the RNA-seq data accurately reflect the abundance of transcript levels.

## DISCUSSION

Transcriptome refers to the complete set of transcripts in a cell, specific to a particular physiological condition or developmental stage (*Wang, Gerstein & Snyder, 2009*). Various technologies have been developed to study the transcriptome, including hybridization-based microarrays, Sanger sequencing-based methods, and RNA sequencing (*Yamada et al., 2003*; *Cheung et al., 2008*; *Wang, Gerstein & Snyder, 2009*). RNA sequencing, also known as RNA-Seq, utilizes the Next Generation Sequencing (NGS) platform and has become a powerful tool for identifying differentially expressed genes (DEGs) and potential molecular mechanisms (*Vogel et al., 2016*; *Wei et al., 2017*; *Yang et al., 2019*). In recent

years, RNA-Seq has been extensively applied in comparative transcriptome analyses of green and purple fruit peppers (*Liu et al., 2020b*; *Tang et al., 2020*; *Meng et al., 2022*).

Unfortunately, RNA-seq often yields a large number of differentially expressed genes (DEGs), making it challenging to identify the specific genes that control a particular phenotype. In this study, we aimed to address this issue by comparing the transcriptomes of three purple-fruited pepper varieties (HN191, HN192, and HN005) with that of one green-fruited variety (EJT) at 30 days after anthesis (DAA). Additionally, we compared the transcriptomes of HN191 and EJT at two earlier developmental stages (10 and 20 DAA). Hence, in total, we conducted five comparisons between green and purple fruit. From these comparisons, we identified a set of 503 common DEGs.

To further narrow down the list of candidate genes, we also utilized transcriptome data from the ripe stage (60 DAA) of four varieties. This selection was based on the understanding that anthocyanins tend to degrade in ripe fruits, leading to a decrease in the expression levels of genes involved in anthocyanin biosynthesis (*Aza-González et al., 2013*; *Borovsky et al., 2004*). By focusing on this stage, we aimed to significantly reduce the number of candidate genes.

Among the 503 common DEGs, we identified 7 structural genes involved in anthocyanin biosynthesis. These genes included *CHS* (T459_15321), *F3H* (T459_06228), *F3′5′H* (T459_28730), *DFR* (T459_06365), *ANS* (T459_00408), *UFGT* (T459_26178), and a novel candidate gene designated as *3RT* (MSTRG.12525). Another structural gene, *CHI* (T459_15095), was not grouped in the 503 common DEGs because its fold change between HN191 and EJT at 10 DAA did not surpass the threshold of 2. The expression levels of these structural genes were relatively low in green fruits (specifically, EJT at 10, 20, and 30 DAA), but exhibited higher expression in purple fruits (specifically, HN191 at 10, 20, and 30 DAA; HN192 and HN005 at 30 DAA), which is consistent with previous studies (*Tang et al., 2020*; *Jung et al., 2019*). The differential gene expression of the structural genes, especially LBGs, is the reason for anthocyanin production (*Tang et al., 2020*). Therefore, these findings suggest that the upregulation of these structural genes is responsible for anthocyanin biosynthesis in purple pepper fruits.

Notably, MSTRG.12525 was identified as a novel structural gene involved in anthocyanin biosynthesis in this study. Blast analysis indicated that the MSTRG.12525 gene shares 99.68% sequence identity with *Capsicum annuum* anthocyanidin-3-O-glucoside rhamnosyltransferase (LOC107861697), which is known to control the conversion of anthocyanidin-3-O-glucosides to anthocyanidin-3-O-rutinosides. Previous research has shown that the predominant anthocyanins accumulated in purple pepper fruits are delphinidin 3-O-rutinoside, delphinidin 3-p-coumaroyl-rutinoside-5-glucoside, and delphinidin 3-cis-coumaroylrutinoside-5-glucoside (*Meng et al., 2022*; *Tang et al., 2020*). Thus, it could be inferred that MSTRG.12525 might play a role in the conversion of 3-O-glucoside to 3-O-rutinoside in purple pepper fruits.

Among the 503 common DEGs, a total of 24 transcription factors were identified, seven of which exhibited significantly higher expression in purple fruits (specifically, HN191 at 10, 20, and 30 DAA; HN192 and HN005 at 30 DAA) than in green fruits. However, their expression levels decreased in ripe fruits (60 DAA), which correlates with the pattern of

anthocyanin accumulation. These findings, suggest that these TFs may play a role in the biosynthesis of anthocyanin.

For gene T459_08508, the expression correlation coefficient between RNA-seq and qRT-PCR was −0.02. As for the gene T459_13190, it encodes a bHLH transcription factor. Its expression level was higher in the unripe fruit of HN191 compared to EJT. However, its expression levels were relatively low in the unripe fruit of HN192 and HN005 at 30 DAA. Therefore, based on these observations, it is likely that these two genes may not be considered as candidate genes.

For the remaining five genes, two of them have already been reported. T459_25692 encodes an MYB transcription factor that is homologous to the *A* gene (GenBank accession number AJ608992.1). It is also known as *CaAN2* since it encodes a homolog of *Petunia anthocyanin 2* (*An2*) (*Borovsky et al., 2004*). Its expression has been detected throughout all stages of fruit development in the purple-fruit variety 5226, but not in the green-fruit variety PI 159234 (*Borovsky et al., 2004*). Overexpression of *CaAn2* in *Nicotiana benthamiana* and *Arabidopsis thaliana* resulted in the development of purple tissues (*Jung et al., 2019*). The promoter of *CaAn2* in purple C. annuum 'KC00134'plants contains an insertion of a non-long terminal repeat (LTR) retrotransposon called CanLTR-A, which may activate the expression of *CaAn2* by recruiting transcription factors at the 3′UTR (*Jung et al., 2019*). On the other hand, T459_03543 encodes a bHLH transcription factor that is homologous to AN1 (LOC107842687). This transcription factor, referred to as CaAN1, along with two other MYBs (CaANT1, LOC107854818; CaANT2, LOC107844901), were significantly upregulated in the purple pepper fruits (*Tang et al., 2020*). They can form an MBW complex with a WD40 protein (CaTTG1, LOC107862994), which is involved in the regulation of structural genes in the anthocyanin biosynthetic pathway (*Tang et al., 2020*).

However, the function of another three TFs in anthocyanin biosynthesis of pepper has not yet been reported. T459_25295 encodes a DIVARICATA-like (DIV) transcription factor, which belongs to the MYB-related family subgroup. Previous studies in chilli pepper (*CaDIV1*, *CaDIV3*, *CaDIV11*) have shown a positive correlation between DIVs and key genes of the flavonoid biosynthetic pathway (*Arce-Rodríguez, Martínez & Ochoa-Alejo, 2021*; *Song et al., 2023*). T459_06113 encodes a bZIP transcription factor. In pepper, a bZIP transcription factor CaHY5 has been found to play a crucial role in regulating anthocyanin accumulation in the hypocotyl. It achieves this by binding to the promoters of key genes involved in anthocyanin biosynthesis, including CaF3H, CaF3'5'H, CaDFR, CaANS, and CaGST (*Chen et al., 2022*). Additionally, research has shown that CaMYB113 can interact with CabHLH143 and CaHY5, suggesting a collaborative role of these three genes in UV-B-induced anthocyanin biosynthesis in pepper fruit (*Wang et al., 2022*). In tomato, a similar bZIP transcription factor called SlHY5 has been found to directly recognize and bind to the promoters of anthocyanin biosynthesis genes, regulating the accumulation of anthocyanin (*Liu et al., 2018a*). Another bZIP transcription factor, MdbZIP44, in apple has been found to promote anthocyanin accumulation in response to ABA by enhancing the binding of MdMYB1 to the promoters of downstream target genes (*An et al., 2018*). T459_26036 encodes a WRKY transcription factor. Previous studies have reported that WRKY transcription factors can interact with MYB and promote anthocyanin biosynthesis

in apple, pear, potato, and eggplant (*An et al., 2019*; *Cong et al., 2021*; *Zhang et al., 2021*; *He et al., 2021*). In summary, these three TFs may represent novel candidate genes that contribute to anthocyanin biosynthesis in pepper fruit.

## CONCLUSIONS

In this study, we investigated the transcriptome dynamics of three purple-fruited varieties (HN191, HN192, HN005) and one green-fruited variety (EJT) during fruit development using RNA-seq. A total of five comparisons between green and purple fruit have been analyzed and 503 common differentially expressed genes (DEGs) have been identified. Among these DEGs, eight structural genes involved in the anthocyanin biosynthesis pathway and 24 transcription factors (TFs) were identified. We hypothesize that one structural gene and three TFs could potentially be novel candidate genes for anthocyanin biosynthesis based on their expression patterns. These findings contribute to a comprehensive understanding of the mechanisms underlying anthocyanin biosynthesis in pepper and provide potential target genes for the genetic improvement of anthocyanin-rich pepper germplasm.

## ACKNOWLEDGEMENTS

We thank the China National GeneBank (CNGB) for providing technical help.

### Funding

This work was supported by the Sichuan Vegetable Innovation Team—Research and Application of Grafting Technology (sccxtd2019-2023), the Tianfu Scholar Program of Sichuan Province (Wei Deng, Department of Human Resources and Social Security of Sichuan Province, 2021-58), and the 14th Five-Year Plan of Vegetable Breeding Project in Sichuan Province—Breeding and Application of new purple pepper germplasm (2021YFYZ0022). The funders had no role in study design, data collection and analysis, decision to publish, or preparation of the manuscript.

### Grant Disclosures

The following grant information was disclosed by the authors:
The Sichuan Vegetable Innovation Team—Research and Application of Grafting Technology: sccxtd2019-2023.
The Tianfu Scholar Program of Sichuan Province (Wei Deng, Department of Human Resources and Social Security of Sichuan Province, 2021-58).
The 14th Five-Year Plan of Vegetable Breeding Project in Sichuan Province—Breeding and Application of new purple pepper germplasm: 2021YFYZ0022.

### Competing Interests

The authors declare there are no competing interests.
## Author Contributions

- Huaqiang Tan conceived and designed the experiments, performed the experiments, analyzed the data, prepared figures and/or tables, and approved the final draft.
- Liping Li conceived and designed the experiments, performed the experiments, prepared figures and/or tables, and approved the final draft.
- Manman Tie performed the experiments, authored or reviewed drafts of the article, and approved the final draft.
- Ronghai Lu analyzed the data, authored or reviewed drafts of the article, and approved the final draft.
- Shaokun Pan analyzed the data, authored or reviewed drafts of the article, and approved the final draft.
- Youwan Tang conceived and designed the experiments, prepared figures and/or tables, and approved the final draft.

## Data Availability

The data is available at the CNGB Sequence Archive (CNSA) of the China National GeneBank DataBase (CNGBdb): CNP0004460.

## Supplemental Information

Supplemental information for this article can be found online at http://dx.doi.org/10.7717/peerj.16792#supplemental-information.

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
