# Peer review of "Transcriptome analysis of green and purple fruited pepper provides insight into novel regulatory genes in anthocyanin biosynthesis"

_PeerJ, doi:10.7717/peerj.16792_

## Round 0.1 · original submission · Major Revisions

The manuscript was reviewed by three independent experts in the field. All the reviewers found the work interesting but raised several issues particularly reviewer 1 and 3 who has valid concerns. The reviewers provide detailed comments in their reviews and point out the areas where the manuscript needs to be improved.

·

Basic reporting

1. Line 152: The brand and model of the equipment used are described, but it would be better to clarify what type of equipment it is to make the reading easier for a more general audience.
2. Line 399: The authors should be more cautious in expressing their findings. Instead of stating that the three TF are novel genes, it would be better to talk about them as novel candidate genes.
3. Line 406: There is a space missing at the beginning of the sentence, before “Among”.
4. Line 406: The word eight is separated between lines 406 and 407.
5. There is certain terminology that needs to be italicized—for example, Capsicum annuum in line 103, Nicotiana benthamiana and Arabidopsis thaliana in line 372.

Experimental design

1. One critical aspect is the selection of purple and green varieties for the study. It is not clear why these specific varieties were chosen over others, and the authors should provide a rationale for their selection. Additionally, only one green-fruited variety was chosen, and it would be helpful to explain the logic behind this decision.
2. Another issue is the difference in the number of time points analyzed for each plate. HN191 was analyzed at three-time points, while HN192 and HN005 were only analyzed at one time point. The authors should explain why this difference in sampling occurred.HN192 and HN005 were harvested at 30 and 60 DAA, why not at 10 and 20 DAA as well?
3. Additionally, HN192 and HN005 were harvested at 30 and 60 DAA, respectively, but not at 10 and 20 DAA. The authors should clarify why these specific time points were chosen.
4. Line 199: The authors should also include a graph showing the results of HPLC analysis, including the control as a reference to confirm that delphinidin was the actual component identified.
5. Line 337: Could the authors further explain their conclusion? It seems to be a correlation between two factors that do not necessarily mean causation.
6. In the introduction, the authors do a nice job briefing the background and importance of anthocyanin, however, they fail to make a comparison with other crops that would emphasize the importance of studying C. annuum over other crops that produce this component as well. Explaining this could add value to the findings.
7. Why HN191 was chosen over the other varieties for microscope observation? What were the criteria for the election?

Validity of the findings

no comment

Additional comments

I would like to highly a strength authors have: S2 table is very informative and thorough, and reflects the hard work done to fulfill the quality of the research. I think it will also be appreciated and welcomed by readers on how to improve their performance and organization in the lab.

Reviewer 2 ·

Basic reporting

Tan et al. performed the transcriptome analysis between green and purple fruited peppers. They identified several potential novel genes involved in anthocyanin pathway in pepper. The manuscript was well written, and the phenotype is very interesting.

I have a few minor concerns as follows:

1. Please provide the full names for all abbreviations, such as in line 89, 102, and 108.

2. In line 29, please provide the full name of DAA here rather than in line 141.

3. In line 199, please provide the methods and results for HPLC analysis.

4. For Fig.5A, please explain what the different colors mean in the legend.

5. In line 298, please provide the methods of the correlation coefficient.

6. In line 348-349, please provide further clarification on how you arrived at this conclusion?

7. In line 351-356, please avoid excessive repetition of the results section in the discussion.

8. Please provide detailed legends for all your supplementary tables. The current versions of your supplementary tables are confusing.

9. In addition to the red color, you mentioned that anthocyanin is also responsible for the red color. At DAA 60, all varieties appeared red, but with reduced anthocyanin. I suggest you discuss this in the discussion.

Experimental design

NA

Validity of the findings

NA

Reviewer 3 ·

Basic reporting

This article basically meets the requirements of the journal, but there are still some formatting problems, such as the scientific name of pepper should be set in italics. In addition, the writing format of the references also needs to be further modified.

Experimental design

The current state of knowledge regarding the anthocyanin biosynthetic pathway and regulatory network is well-established. In this study, transcriptomic analysis was performed by sampling fruits at various developmental stages. This approach provided an initial analysis of the gene expression patterns related to anthocyanin biosynthesis. This can be regarded as an innovative aspect of this article. However, no functional analysis was conducted on the differentially expressed genes. Moreover, there was an insufficient amount of experimental data to validate the findings of transcriptomic data analysis.

Validity of the findings

The content of the article needs further improvement due to several inappropriate descriptions. For instance, despite the localization and functional analysis of HY5 in chili peppers, the article incorrectly characterizes it as a transcription factor of unknown functions in this species.

Additional comments

The charts and figures can be refined to enhance their visual appeal.

---

## Round 0.2 · Minor Revisions

Although authors have addressed the comments raised during the review process, however manuscript still needs a revision for fixing minor concerns pointed out by the reviewer 3.

·

Basic reporting

The authors have taken all the initial concerns into account and have made the necessary adjustments in the manuscript's details, as well as providing a comprehensive response to the reviewers' comments. Based on these changes, I am confident that the manuscript is now ready for submission and no further revisions are necessary.

Experimental design

The previous concerns have been adequately resolved, and there is no need for any further clarifications.

Validity of the findings

The authors have taken all the initial concerns into account and have made the necessary adjustments in the manuscript's details, as well as providing a comprehensive response to the reviewers' comments. Based on these changes, I am confident that the manuscript is now ready for submission and no further revisions are necessary.

Reviewer 3 ·

Basic reporting

Line 227: There is a formatting error,byte units should be capitalized,such as "GB".
Line 286: The names of genes should be italicized.
Line 288: There should be spaces between English words and numbers.

Experimental design

No comment.

Validity of the findings

No comment.

Additional comments

No comment.

---

## Round 0.3 · accepted · Accept

The manuscript is revised by the authors satisfactorily and is ready for publication in its current form.